# Human Pluripotent Stem Cell–Mesenchymal Stem Cell-Derived Exosomes Promote Ovarian Granulosa Cell Proliferation and Attenuate Cell Apoptosis Induced by Cyclophosphamide in a POI-like Mouse Model

**DOI:** 10.3390/molecules28052112

**Published:** 2023-02-24

**Authors:** Lifan Zhang, Yabo Ma, Xianguo Xie, Changzheng Du, Yan Zhang, Shaogang Qin, Jinrui Xu, Chao Wang, Yi Yang, Guoliang Xia

**Affiliations:** 1Key Laboratory of Ministry of Education for Conservation and Utilization of Special Biological Resources in the Western, Ningxia University, Yinchuan 750021, China; 2School of Life Sciences, Ningxia University, Yinchuan 750021, China; 3State Key Laboratory of Agrobiotechnology, College of Biological Sciences, China Agricultural University, Beijing 100193, China

**Keywords:** hiMSC exosomes, premature ovarian insufficiency, ovarian granulosa cell, cell proliferation, cell apoptosis

## Abstract

Premature ovarian insufficiency (POI) is a complex disease which causes amenorrhea, hypergonadotropism and infertility in patients no more than 40 years old. Recently, several studies have reported that exosomes have the potential to protect ovarian function using a POI-like mouse model induced by chemotherapy drugs. In this study, the therapeutic potential of exosomes derived from human pluripotent stem cell–mesenchymal stem cells (hiMSC exosomes) was evaluated through a cyclophosphamide (CTX)-induced POI-like mouse model. POI-like pathological changes in mice were determined by serum sex-hormones levels and the available number of ovarian follicles. The expression levels of cellular proliferation proteins and apoptosis-related proteins in mouse ovarian granulosa cells were measured using immunofluorescence, immunohistochemistry and Western blotting. Notably, a positive effect on the preservation of ovarian function was evidenced, since the loss of follicles in the POI-like mouse ovaries was slowed. Additionally, hiMSC exosomes not only restored the levels of serum sex hormones, but also significantly promoted the proliferation of granulosa cells and inhibited cell apoptosis. The current study suggests that the administration of hiMSC exosomes in the ovaries can preserve female-mouse fertility.

## 1. Introduction

Premature ovarian insufficiency (POI) refers to physical amenorrhea in women before the age of 40. POI is characterized by hypoestrogenism, infertility and, most importantly, a reduction in the follicle pool [1]. An epidemiological investigation showed that the incidence of POI is 1% in women under 40 years of age [2]. However, the causative reasons for POI remain elusive. Besides gene mutation, the incidences of either autoimmunity, vaccination, tumor therapy or environmental factors are all responsible for POI [3]. Unfortunately, a large portion (about 37%) of POI in young women is caused by chemotherapy for cancer [4]. Some chemotherapy drugs include alkylating agents which have been shown to be gonadotoxic [5]; CTX is not only one of the most widely utilized antineoplastic drugs, but also an effective immunosuppressant and the most commonly used drug in blood and marrow transplantation. Unfortunately, the administration of CTX results in DNA double-strand breakdown (DSB) and leads to cell death [6]. Several studies report that CTX induces ovarian granulosa cell apoptosis and, consequently, results in loss of follicles both in vitro and vivo [7,8,9]. However, there is a lack of effective measures to alleviate CTX-induced POI in the clinical setting.

Hormone replacement therapy is the most commonly used treatment for POI in the clinical setting, which is accompanied by side effects [10,11,12]. Therefore, it is urgent to explore and adapt new treatments for POI. Alternatively, human mesenchymal stem-cell (hMSC) therapy is a promising strategy to restore ovarian function and cure female infertility [13]. The available hMSCs include human bone-marrow mesenchymal stem cells [14], human adipose-derived stem cells [15] and human amniotic mesenchymal stem cells [16]. Notably, human pluripotent stem cells (hPSCs) have been proven to be effective for preventing tumor expansion and the inhibition of metastasis in mice [17]. However, the potential role of hiMSC, which is further differentiated from hPSC when protecting ovaries from chemical toxins, is unknown.

Recent studies have applied exosomes for medical use. Exosomes are vesicles with a size of 30 nm–150 nm secreted by MSCs [18]. Exosomes are active in modulating both cellular communication and the fate of recipient cells by releasing non-coding RNA, mRNA and proteins [19,20,21]. Accumulating evidence shows that exosomes derived from distinct hMSCs inhibited mouse ovarian-granulosa-cell apoptosis [22,23,24]. However, whether transferring the hiMSC exosomes to the target ovarian cells might recover ovarian function and inhibit granulosa cell apoptosis is not known.

As is known, not only are aging somatic cells prone to oxidative stress, but germ cells are sensitive to redox balance, antioxidant enzymes and oxidase levels [25,26]. When growing follicles, CTX does not directly induce oocyte death, but induces granulosa cell apoptosis [27]. CTX causes high-level oxidative stress in human granulosa cells and results in apoptosis [28]. Notably, as the central redox sensor in the cells, nuclear factor erythroid 2-related factor 2 (NRF2) translocates into the nucleus, binds to the AU-rich element, and promotes the expression levels of downstream related antioxidant enzyme genes under oxidative stress [29]. We, therefore, wonder if NRF2 and related proteins will be affected by CTX and if hiMSC exosomes will contribute to alleviating the oxidative stress induced by CTX.

This study was based on a POI-like mouse model induced by CTX to evaluate the effects of hiMSC exosomes on mouse ovarian function, granulosa cell proliferation and the expression levels of proteins related to apoptosis and oxidative stress. 

## 2. Results

### 2.1. Identification of hiMSCs

In order to ensure that hiMSCs are a type of MSC, the surface markers and differentiation ability of hiMSCs were examined. The cell morphology of the fifth passage of hiMSCs after 3 days of culture was observed under a light microscope (Figure 1A). Alizarin Red staining and Oil red O dyeing proved that hiMSCs were able to differentiate into either adipocytes or osteocytes in response to the differentiation ability, as expected (Figure 1B,C), which implies that the developmental potential of the cultured hiMSCs was functional. Additionally, flow cytometry (FC) results revealed that the hiMSCs expressed positive mesenchymal stem-cell markers such as CD73, CD90 and CD105, while the hematopoietic cell surface markers, such as CD11b, CD14, CD34, CD45 and HLA-DR, were shown to be negative in hiMSCs (Figure 1B).

### 2.2. Collection of Exosomes from hiMSCs

Recent studies have reported that exosomes are secreted from the cultural supernatant of MSCs. In order to collect hiMSCs after the collection of exosomes, the characteristics of nanoparticles derived from hiMSCs were evaluated through assays, including Western blotting, FC, transmission electron microscopy (TEM) and nanoparticle tracking analysis (NTA). The results demonstrated that characteristic hiMSC exosomes were obtained as the exosomal markers, such as TSG101 and FLOTILLIN-1, were positively expressed, while other intracellularly expressed proteins, such as CANEXIN and GAPDH, were negatively expressed (Figure 2A). According to the FC assay, exosomal surface markers such as CD9 and CD63 were positively expressed in hiMSC exosomes (Figure 2B). Furthermore, TEM results showed that hiMSC exosomes were 30–150 nm spheres and exhibited a complete membrane structure (Figure 2C). Finally, NTA results showed that the mean diameter of the hiMSC-exosomes was 124.4 nm (Figure 2D). These data demonstrated that the recovered matter from the cultured hiMSCs were hiMSC exosomes and were available for future therapeutic experiments.

### 2.3. A POI-Like Mouse Model Was Established by CTX

A POI-like mouse model was induced by CTX as reported previously. To ensure that the CTX-induced POI-like mouse model was successfully established, the body weight of the mice was measured every 3 days during the modeling establishment. The results demonstrated that the body weight of the CTX-group mice gradually reduced compared to that in the control group (Figure 3A). Meanwhile, the ovarian size and weight in the CTX group reduced dramatically as compared to the control group (Figure 3B,C). ELISA results showed that the level of follicle-stimulating hormone (FSH) in the CTX group was obviously higher than that in the control group (*p* < 0.05), while the levels of estradiol (E_2_) and anti-Mullerian hormone (AMH) in the CTX group were significantly lower than those in the control group (*p* < 0.01) (Figure 3D). Additionally, vaginal-smear results showed that the regular estrous cycle was approximately 5 days in the control group, but it was disturbed in the POI-like mouse model, in which most of the POI-like mice stayed in the estrous phase (Figure 3E). Over the course of the 4-month mating trial, it was observed that, after CTX treatment, the mating period of mice was prolonged to an average of 4 weeks. Additionally, the number of pups produced by mice in the CTX group was dramatically reduced compared to the control group (Figure 3F). Hematoxylin staining results also revealed that the number of both the primordial follicles and the growing follicles in the CTX group were dramatically reduced compared to the control group (*p* < 0.001) (Figure 3G). These results proved that the CTX-induced POI-like mouse model was successfully established.

### 2.4. Determination of Time Point for Exosomes Treat in POI-Like Mice 

To determine the effect of CTX on follicular reserve, mice ovaries were hematoxylin- stained. While on the fourth day of modeling the number of each stage of follicles in the CTX group dropped compared to the control group, it was not statistically significant (*p* > 0.05) (Figure 4A). On the fifth day, the number of the primordial follicles significantly dropped compared to the control group: a decrease by 40.32% (*p* < 0.05). Meanwhile, the total number of follicles also decreased significantly (*p* < 0.05) (Figure 4B). On the sixth day, the number of both primordial follicles and primary follicles were significantly reduced in the CTX group as compared to the control group (*p* < 0.001) (Figure 4C). These data demonstrate that CTX accelerated the loss of primordial follicles from the fifth day onwards, providing an accurate time measurement for hiMSC exosome treatment.

### 2.5. hiMSC Exosomes Restored Ovarian Function in POI-Like Mouse Model

The effect of hiMSC exosomes on the POI-like mouse model was unknown. hiMSC exosomes were in-situ injected into the ovarian bursa of POI-like mice on the fifth day, and the therapeutic effect of hiMSC exosomes in terms of alleviating POI was evaluated (Figure 5A). The results revealed that treatment of hiMSC exosomes did not cause obvious body-weight change between the CTX group and the CTX + exosome group (Figure 5B). However, the ovarian weight in the CTX group significantly reduced as compared to the CTX + exosome group (Figure 5C). In addition, ELISA results revealed that the level of FSH (23.06%) was sightly decreased in response to hiMSC exosome treatment and the levels of both E_2_ (68.43%) and AMH (83.21%) were significantly increased in the CTX + exosome group compared to the CTX group (Figure 5D). Additionally, hiMSC exosomes administration also resulted in a reduction in follicle loss at any stage, particularly the primordial follicles in the CTX + exosome group, as compared to the CTX group (Figure 5E). Hence, hiMSC exosomes could restore the impaired ovarian function and ovarian reserve induced by CTX.

### 2.6. hiMSC Exosomes Promoted Ovarian Granulosa Cell Proliferation and Simultaneously Inhibited Cell Apoptosis 

To evaluate the effect of hiMSC exosomes on granulosa cell proliferation, the following results are provided. Briefly, the immunohistochemistry results demonstrated that the level of the kI67 protein in the antral follicles of the CTX group reduced dramatically (58.3%) when compared to the CTX + exosome group (Figure 6A). Additionally, the immunofluorescence results indicated that PCNA expression levels in the antral follicles of the CTX + exosome group were higher than those in the CTX group (25.6%) (Figure 6B). Additionally, there was no difference in the KI67 and PCNA expression levels between the control and exosome groups (Figure 6A,B). Therefore, CTX administration impaired mouse granulosa cell proliferation. Collectively, hiMSC exosome supplementation restored the proliferation capability of granulosa cells after CTX administration.

To verify if CTX induced granulosa cell apoptosis and if hiMSC exosomes were able to alleviate cell death, the cellular apoptotic signals were evaluated. The data demonstrated that the level of CASPASE 3 in the CTX + exosome group was lower than that in the CTX group (Figure 7A). In contrast to the control group, where fewer apoptotic cells were observed, a great number of apoptotic cells were observed in the CTX group. Meanwhile, the number of apoptotic cells in the CTX + exosome group was significantly reduced (36.4%) compared to those found in the CTX group (Figure 7B). These results proved that hiMSC exosomes can inhibit granulosa cell apoptosis in mice treated with CTX.

### 2.7. hiMSC Exosomes Can Alleviate Oxidative Stress Produced in Mouse Ovarian Granulosa Cells 

To verify if hiMSC exosomes could effectively prevent the oxidative stress induced by CTX in granulosa cells, the following assays were performed. The MTT assay showed that 50 μM 4-hydroxycyclophosphamide (4–OHCP) significantly reduced the survival rate of ovarian granulosa cells. However, when 1μg of hiMSC exosomes was applied to the cells, the level of cellular survival rate increased significantly (Figure 8A). In addition, the protein expression levels of both BAX and cleaved CASPASE 3 were downregulated and the expression level of Bcl-2 was upregulated in the CTX + exosome group as compared to the CTX group (Figure 8B). Further, the expression of the NRF2 protein was dramatically upregulated in the CTX + exosome group compared to the CTX group. In line with this, the expression levels of SOD1 and GCLC were upregulated in the CTX + exosome group compared to the CTX group (Figure 8B). Collectively, hiMSC exosomes can upregulate the expression of NRF2 in granulosa cells, thus promoting the expression of oxidative-stress-related proteins, inhibiting cell apoptosis and promoting cell proliferation.

## 3. Discussion

The most specific characteristics of POI are follicle dysfunction and follicle depletion [30]. Importantly, the occurrence of POI is closely associated with the quality of ovarian granulosa cells [30]. As has been shown, the toxicity of CTX is not directly exerted on oocytes, but on granulosa cells, which induces DNA DSB and apoptosis of granulosa cells in growing follicles [9,31]. Consequently, trying to prevent follicular granulosa cells from apoptosis could be a suitable and practical target in POI treatment. In this study, as one of the most novel candidate tools for cell-free therapy [32], hiMSC exosomes were proven to be effective in alleviating granulosa cell apoptosis through intrabursal administration of hiMSC exosomes to POI-like mice.

Exosomes derived from different cell types play an important role in mediating intercellular communication by transferring proteins, lipids and RNA between cells [33,34], which is also helpful for regulating the fate of ovarian granulosa cells [9,35,36]. Several studies have proven that human-stem-cell-derived exosomes are effective in preventing granulosa cell apoptosis. For instance, not only do exosomes derived from human-umbilical-cord mesenchymal stem cells (hUCMSC exosomes) inhibit ovarian granulosa cell apoptosis in vitro [35], but miR-126-3p containing hUCMSC exosomes also attenuated ovarian granulosa cell apoptosis in a rat model of POI [37]. Further, human-amniotic-fluid stem-cell-derived exosomes carrying miR-369-3p downregulated the expression of YAF2, which attenuates ovarian granulosa cell apoptosis by regulating the PDCD5/P53 signaling pathway [36]. In line with these findings, our study also proved that hiMSC exosomes are a candidate material for the prevention of ovarian granulosa cell apoptosis. We found that hiMSC exosomes upregulated NRF2 protein levels in granulosa cells induced by 4-OHCP in vitro and promoted the expression of oxidative-stress-related proteins, including SOD 1 and GCLC. Knockdown miR-122-5p in ovarian exosomes attenuated the apoptosis of mouse ovarian granulosa cells as well [9]. Collectively, applying exosomes could be useful for preserving follicle atresia by rescuing granulosa cell apoptosis. Despite these findings, however, more in-vivo assays and more specific studies are needed to clarify the specific molecules that are pivotal for supporting granulosa cell life.

Although the mechanism of CTX toxicity on the ovaries has not been fully addressed, it is predicted to increase reactive-oxygen-species (ROS) production and decrease antioxidant activity because of CTX-induced oxidative stress [38]. Conventionally, ROS acts as a signal substance and plays a crucial role in regulating various activities in both the reproduction and development of mammals [39,40]. Overstimulated ROS may damage the molecular structures of either the nucleic acid, lipids or proteins within the cells [41]. For instance, Glutathione Peroxidase 4 (GPX4) can convert lipid peroxides into non-toxic lipophilic alcohol forms [42]. Further, glutathione synthetase (GSS) and glutamate cysteine ligase catalytic subunit (GCLC) promote GPX4 or glutathione (GSH) biosynthesis [43]. Alternatively, Cu/Zn Superoxide Dismutase (SOD1) is commonly known for its scavenging activity, which is able to convert superoxide radicals into molecular oxygen and hydrogen peroxide [44]. In agreement with these findings, we have proven that hiMSC exosomes not only upregulate NRF2 protein levels in granulosa cells, but promote the expression of oxidative-stress-related proteins, including SOD 1 and GCLC. Collectively, these results might suggest one of the mechanisms of hiMSC exosomes, which promote the expression levels of antioxidant enzymes by upregulating NRF2, thus inhibiting the apoptosis of mouse ovarian granulosa cells. However, we need more evidence to support our conclusion, which can be more clearly elucidated with further experiments.

The efficiency of delivering enough exosomes to the targeted ovaries is another key point for assisting reproduction in the future. Previous studies reported that exosomes derived from various MSCs were injected via the tail vein [36,45,46]. However, the liver and spleen, instead of the ovaries, are the major targets of exosomes injected via the tail vein [47]. Additionally, exosomes derived from different cell types may have distinct tissue-specific homing effects [48]. In this study, the hiMSC exosomes were in-situ injected into the ovarian bursa of POI-like mice to evaluate the effectiveness of the exosomes. According to the data of ovarian weight increase and restored follicle numbers, hiMSC exosome injection may help to restore ovarian function. Indeed, this methodology is only restricted to rodent animal models instead of other mammals because only rodents, such as mice, have an insulated bursa structure outside the ovaries [49]. Interestingly, a recent study has proven that the administration of hUCMSC exosomes through bursa injection contributes to the activation of dormant primordial follicles and facilitates reproduction in a POI mice model [50]. Another issue that needs to be fully studied is the methodology of purifying the exosomes. Although many experiments have been performed, there is no uniformity in the isolation method of exosome purification so far. The present methods mainly include immunogold labeling, continuous sucrose gradient, differential ultracentrifugation and exosome isolation kits [51]. The most commonly used method is ultracentrifugation due to the superlative quality of the exosomes isolated within it and the ubiquity of its usage [51,52]. In conclusion, exploring more specific measures to facilitate hiMSC exosome purification in vitro and its absorption by ovaries are urgently required to further approve the usefulness of stem-cell-derived exosome-oriented medication for POI patients.

## 4. Materials and Methods

### 4.1. hiMSC Culture and Identification 

hiMSCs were purchased from in Nuwacell biotechnologies company (Hefei China). The cells were planted at a density of 5 × 10^6^ cells in hiMSC growth media and were cultured at 37 °C and 5% CO_2_. The morphology of hiMSCs at the 5th generation was observed under a light microscope. hiMSCs were identified by examining the specific surface markers, including CD73, CD90 and CD105 (all from BD, USA). Contrarily, the nonspecific markers, such as CD14b, CD34, CD11b, CD45, and HLA-DR (all from BD, Township of Franklin, USA), were examined using flow cytometry. Cells were cultured for 3 weeks by adding adipogenic differentiation media (RP02014-A, Nuwacell biotechnologies company, Hefei, China) and osteogenic differentiation media (RP02014-C, Nuwacell biotechnologies company, Hefei, China), respectively. Oil Red O and Alizarin Red dye were applied to verify the differentiation ability of hiMSCs. hiMSCs from passage 5 were used for this study.

### 4.2. Preparation of hiMSC Exosomes

In order to collect exosomes from the supernatant of cultured hiMSCs, exosomes were purified by differential centrifugation. Briefly, hiMSCs at passage 5 were in-vitro cultured for 3 days to reach 80% confluence. Then, the culture supernatant was collected. To obtain pure exosomes, the supernatant was centrifuged at different speeds, of 300× *g* for 10 min, 2000× *g* for 10 min and 10,000× *g* at 4 °C for 30 min, to remove sediment and cell debris. Following this, supernatant was centrifuged at 100,000× *g* for 2 h to pellet the exosomes. Next, the sediment was re-suspended in particle-free PBS and centrifuged again at 100,000× *g* for 2 h. Finally, the exosomes were re-suspended in 100 µL of particle-free PBS.

The size distribution and concentration of hiMSC exosomes were measured using NTA by ZetaView PMX 110 (Particle Metrix, Grafelfing, GER). The morphology of hiMSC exosomes was observed by TEM (Hitachi H-7650, Tokyo, JPN. The exosome markers (CD9 and CD63) were detected by FC. Additionally, the expression levels of TSG-101 (Santa Cruz, CA, USA), FOLITILLIN-1 (BD, Township of Franklin, NJ, USA), CALNEXIN (Abcam, Cambridge, UK) and GAPDH (CST, Danvers, CO, USA) were assessed using Western blotting in both hiMSC exosomes and hiMSCs. 

### 4.3. Establishment of POI-Like Mouse Model and In Situ Injection of hiMSC Exosomes into Mice Ovarian Bursa 

C57 BL/6 female mice of 8 weeks of age were purchased from Gempharmatech (Nanjing, China). The utilization of the animals was approved by the Animal Welfare & Ethics Committee of Ningxia University. Intraperitoneal injection of CTX for 14 days (50 mg/kg) was used to establish a POI-like mouse model according to [53], while the control group received a single intraperitoneal injection with an equal amount of PBS. The mice were divided into four groups (*n* = 6 for each group): control group (treatment with PBS), CTX group (treatment with CTX), exosome group (PBS group treatment with hiMSC exosomes) and CTX + exosome group (CTX treatment with hiMSC exosomes). Mice from all groups were bred at a temperature of 25 ± 2 °C with a 12 h light/dark cycle with free access to chow and water. Vaginal smears were taken at 8 a.m. each day to measure the estrous cycle.

### 4.4. Fertility Testing

Mating trials were initiated at 1 week after POI-like mouse-model-establishment cessation. Eight-week-old male mice were used for mating. Briefly, one male mouse was placed in a cage with one female mouse from either the control group or the CTX group for several months (*n* = 5 for each group). The mice were sacrificed at 7 months of age. The number of offspring was recorded per week. The reproductive data were summarized using the reproductive curve. 

### 4.5. ELISA Assay

Mice blood samples were obtained from eyeball veins. To obtain the mice sera, blood samples were centrifuged at 300× *g* at 4 °C for 15 min. The levels of FSH, E_2_ and AMH of mice sera were detected according to the manufacturer’s instructions using by ELISA kits (Cusabio, Wuhan, China).

### 4.6. Hematoxylin Staining and Ovarian Follicle Count

During the POI-like mouse-model establishment, mice ovaries were collected from the control group and CTX group from the 4th day to 6th day (*n* = 4 for each group). Since POI-like mouse was successfully established, the mice were sacrificed at the age of 10 weeks. Mice ovaries were collected and fixed in 4% (*w*/*v*) paraformaldehyde for 24 h. Then, the ovaries were embedded in paraffin after dehydration and, subsequently, serially cut into 8 μm sections. Each section was stained with hematoxylin to count the follicle numbers of primordial, primary, secondary and antral follicles under light microscopy.

### 4.7. Immunohistochemistry

The ovarian tissues were cut into 5 μm thick sections. The sections were dewaxed in xylene and gradient alcohol, and then blocked with 10% donkey serum (ZSGB-BIO, Zhongshan, China). Following this, the slides were incubated with the primary antibodies (Abcam, UK) of either Ki67 or CASPASE 3 overnight at 4 °C. Then, the slides were incubated with respective HRP-labeled secondary antibodies and visualized with diaminobenzidine (DAB). Finally, cell nuclei were stained with hematoxylin. The slides were observed and imaged using a fluorescence microscope (Olympus, Tokyo, JPN).

### 4.8. Immunofluorescence Staining

Each 5 μm thick wax section was blocked with 10% donkey serum and incubated with the primary antibody PCNA (Sant Cruz, Santa Cruz, CA, USA) overnight at 4 °C, which was followed by incubation with Alexa Flour 555-conjugated secondary antibody (Yeasen, Shanghai, China). Finally, the cell nuclei were stained with DAPI (Beyotime, Shanghai, China) for 5 min. The slides were then observed and imaged using DMI 8 (Leica stellaris 5, Wetzlar, GER).

### 4.9. TUNEL Analysis

The sections of mice ovaries were processed using a TUNEL apoptosis detection kit (Vazyme, Nanking, China). Briefly, the slides were incubated with the primary antibody FOXL2 (Novus Biologicals, Denver, CO, USA). Next, the TUNEL detection solution was incubated at 37 °C without light for 1 h. The nuclei were stained with DAPI for 5 min. Finally, the slides were observed and imaged using DMI 8.

### 4.10. Isolation, Culture and Treatments of Mouse Granulosa Cells

The mice were euthanized to collect the ovaries. The follicles were punctured with a 30-gauge needle, and the granulosa cells were extracted under a stereoscopic microscope. Collected cells were washed three times with PBS. DMEM/F12 medium (containing 10% (*v*/*v*) FBS, 100 IU/mL penicillin and 100 mg/mL streptomycin) was used to re-suspend the pellet. Then, the suspensions were added to plates, which were incubated at 37 °C overnight with 5% CO_2_ to allow cells to attach. After culture, cells in 6-well plates were used to obtain target proteins, and cells in 96-well plates were used to detect survival rate, respectively. The solution of 4-OHCP (Medbio, Shanghai, China) was prepared firstly by dissolving in DMSO to 200 mM, followed by diluting to 12.5 μM, 25 μM, 50 μM and 100 μM in DMEM/F12 medium for further experiments.

### 4.11. MTT Assay

About 8000 mouse ovarian granulosa cells were seeded into a 96-well plate for each well. Different concentrations of exosomes were added and co-cultured with 4-OHCP for 24 h. Cell viability was determined by adding 25 μL of 5 mg/mL MTT to each well. After 4 h of incubation at 37 °C, the supernatant was replaced with a 110 μL aliquot of DMSO in the dark. The absorbance at 570 nm was measured using EnSpire (Perkin Elmer, Waltham, MA, USA).

### 4.12. Western Blotting Assay

Mouse ovarian granulosa cells were lysed to extract protein using a lysis buffer (KeyGen, Nanking, China). The protein concentration was measured using a BCA protein assay kit (KeyGen, China). The protein was separated using SDS-PAGE and then transferred onto a PVDF membrane (Millipore, Billerica, MA, USA). Next, the membrane was blocked with 5% skim milk for 1 h and then incubated with the primary antibody at 4 °C overnight. Following this, membranes were incubated with HRP-conjugated goat anti-rabbit or goat anti-mouse secondary antibodies (1:10,000, ZSGB-BIO, Zhongshan, China) at room temperature for 1 h. The ECL (Thermo Fisher Scientific, Wilmington, MA, USA) chemiluminescence reagent was prepared and developed. Finally, the proteins were detected using an Amersham Imager 600 (GE, Boston, MA, USA).

### 4.13. Statistical Analysis

GraphPad Prism 8 software was used for statistical analysis. The results are presented as the mean ± SD. One-way analysis of variance (ANOVA) and Student’s *t*-tests were used for statistical comparisons among different groups; *p* < 0.05 was considered to be significantly different. Each experiment was performed at least three times.

## 5. Conclusions

Premature-ovarian-failure patients face irreversible ovarian damage. Although stem cells have achieved some excellent results in clinical settings, exosomes are becoming increasingly important in the field of regenerative medicine. Exosomes can be directly isolated from cell cultures and carry no risk of tumor formation. In the future, exosome therapy has the potential to become an ideal treatment approach. This study proves that hiMSC exosomes contribute to alleviating the granulosa cell apoptosis induced by CTX in POI-like mouse ovaries. The elevated anti-oxidative response aroused by hiMSC exosomes could be a major contribution towards preventing granulosa cell apoptosis.

## Figures and Tables

**Figure 1 molecules-28-02112-f001:**
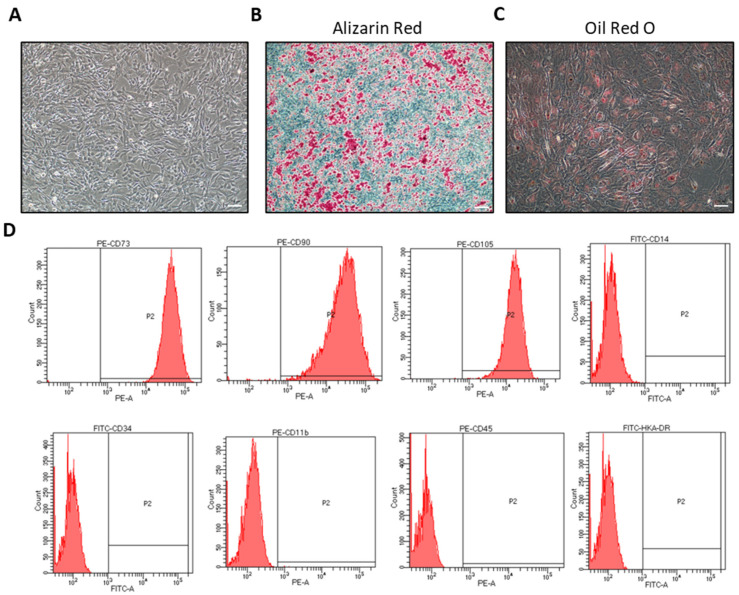
Confirmation of hiMSC characteristics. (**A**) The fifth passage of hiMSCs cultured in vitro for 3 days was observed under a microscope. (**B**) Osteoblasts were identified by Alizarin Red staining, in which a red stain showed cells with calcium accumulation. (**C**) Adipogenesis was identified by Oil Red O staining, in which red satin showed accumulation of neutral lipid within the cells. (**D**) Phenotypes of hiMSCs were detected by FC. hiMSCs expressed CD73, CD90 and CD105 highly. Contrarily, the expression of CD14b, CD34, CD11b, CD45 and HLA-DR was negative in the hiMSCs. Scale bar: 200 μm.

**Figure 2 molecules-28-02112-f002:**
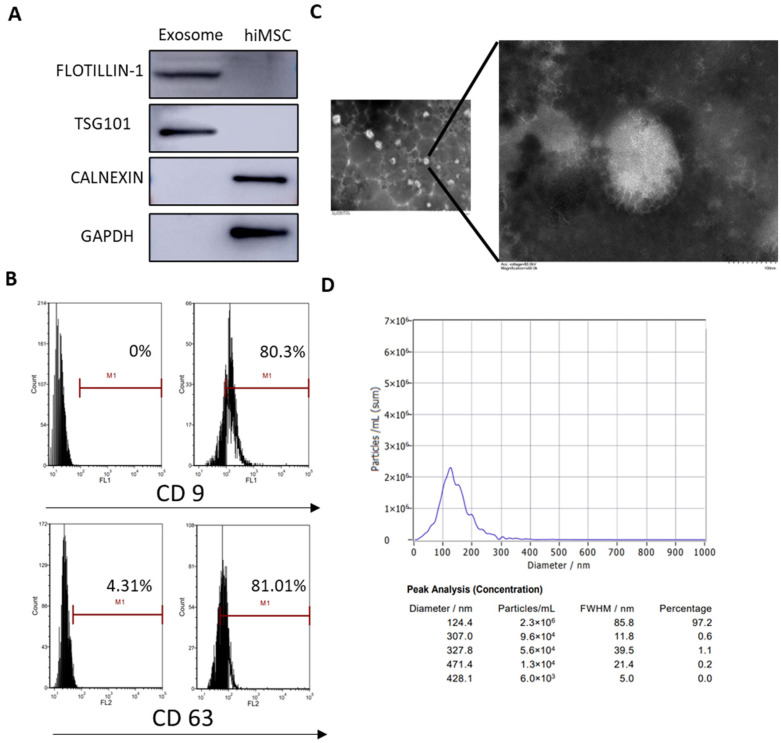
Characterization of hiMSC exosomes. (**A**) Western blotting was applied to analyze the expression of FLOTILLIN-1, TSG101, CALNEXIN and GAPDH in hiMSC exosomes. (**B**) FC was employed to detect the expression of exosomal surface markers such as CD9 and CD63. (**C**) TEM was performed to observe the morphology of hiMSC exosomes. (**D**) NTA was used to measure the size distribution and concentration of hiMSC exosomes. The scale bars represent 1 μm and 100 nm.

**Figure 3 molecules-28-02112-f003:**
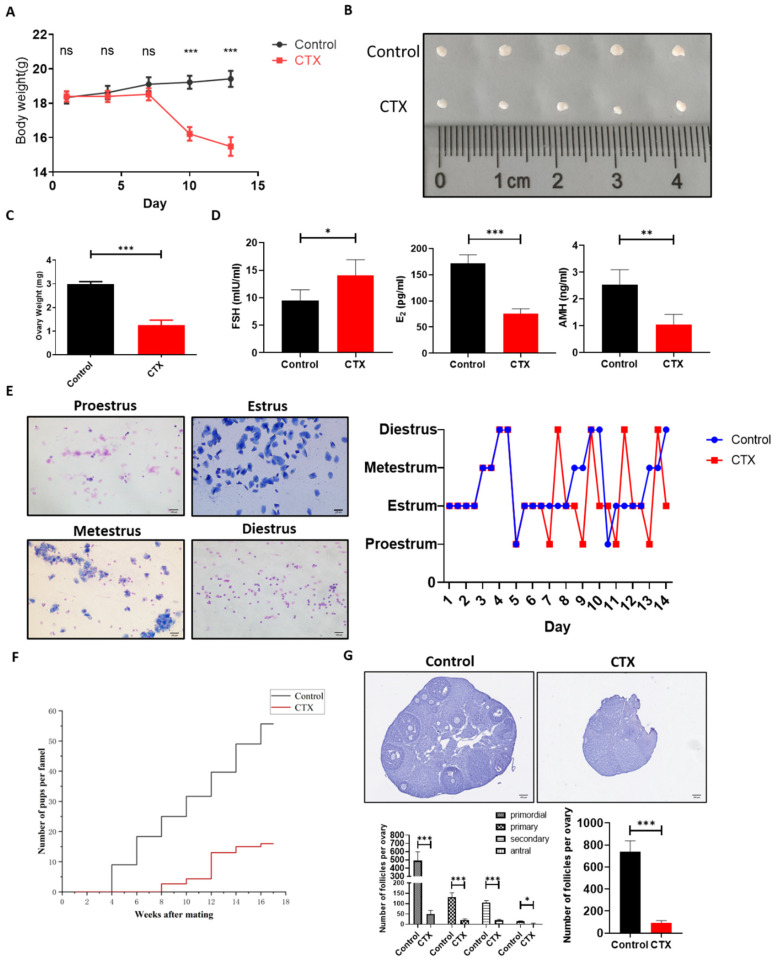
Establishment of POI-like mouse model. (**A**) The weight of mice administered with CTX gradually decreased as compared to the control group. (**B**,**C**) The ovarian size and weight of POI-like mice decreased dramatically. (**D**) The level of FSH in the POI-like mice sera was significantly higher than those in the control group. The levels of E_2_ and AMH significantly decreased with administration of CTX to the mice. (**E**) The estrous cycle of mice in the CTX group began to be disordered on day 5. (**F**) Distribution of offspring in the control and CTX group. Mouse fertility in the CTX group was impaired and fewer mice were born as compared to those in the control group. (**G**) The number of primordial follicles and growing follicles were counted on the 14th day of establishment of the POI-like mouse model (*n* = 5). The scale bars represent 100 μm (**E**) and 200 μm (**G**), respectively. Data are presented as the mean ± SD, * *p* < 0.05, ** *p* < 0.01, *** *p* < 0.001; ns, no significance.

**Figure 4 molecules-28-02112-f004:**
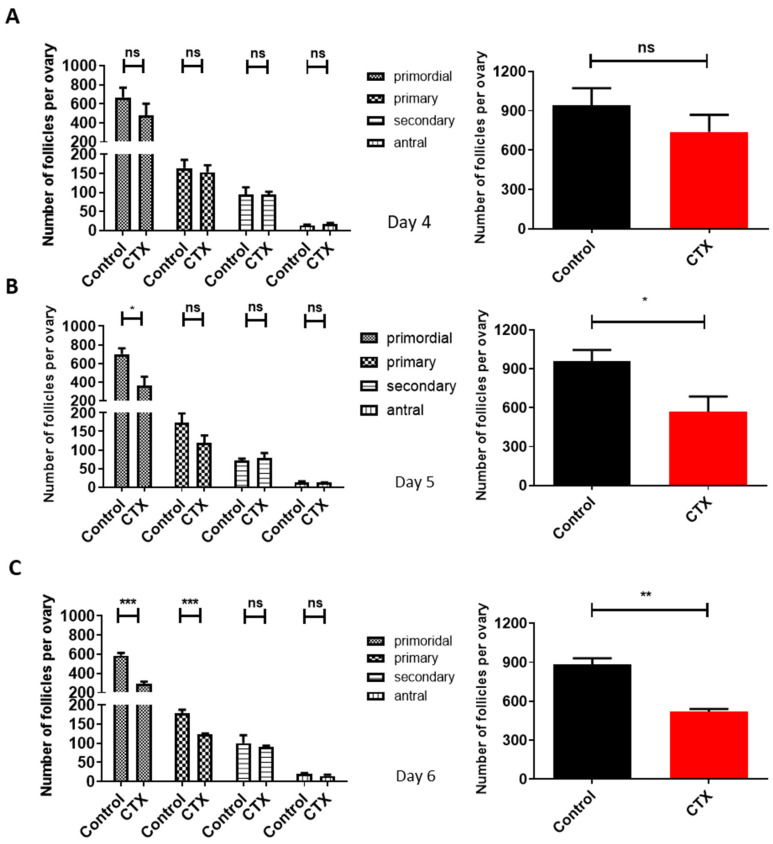
Determination of time points of hiMSC exosome treatment in POI-like mice. (**A**–**C**) The number of the primordial follicles and growth follicles were counted on the fourth, fifth and sixth days of the established POI-like mouse model. From the fifth day on, the number of primordial follicles and total follicles significantly decreased following CTX administration (*n* = 4). * *p* < 0.05, ** *p* < 0.01, *** *p* < 0.001; ns, no significance.

**Figure 5 molecules-28-02112-f005:**
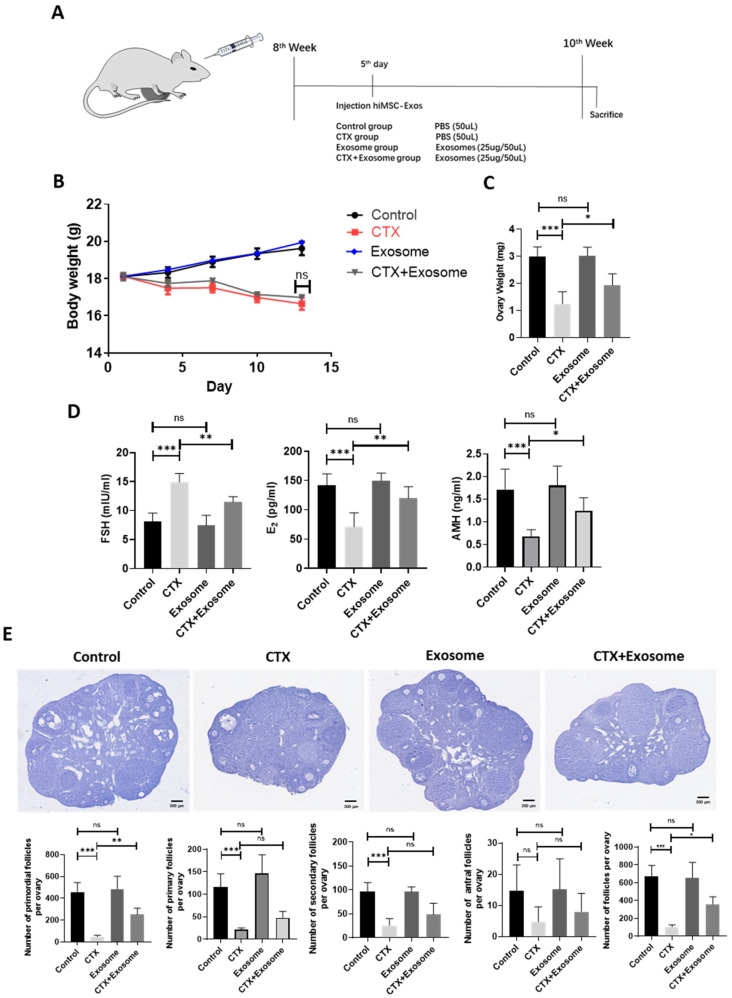
hiMSC exosomes restore the impaired mouse follicle function and follicle reserve induced by CTX administration. (**A**) CTX treatment strategy for producing POI-like mouse model. (**B**) The body weight in the CTX + exosome group was similar to that of the CTX group. (**C**) The ovarian weight in the CTX + exosome group were obviously increased compared to the CTX group. (**D**) The level of FSH in the CTX + exosome group was significantly decreased, while the levels of both E_2_ and AMH were remarkably increased, as compared to the CTX group. (**E**) The loss of primordial follicles and total follicles was alleviated in the CTX + exosome group compared to the CTX group (*n* = 6). * *p* < 0.05, ** *p* < 0.01, *** *p* < 0.001; ns, no significance.

**Figure 6 molecules-28-02112-f006:**
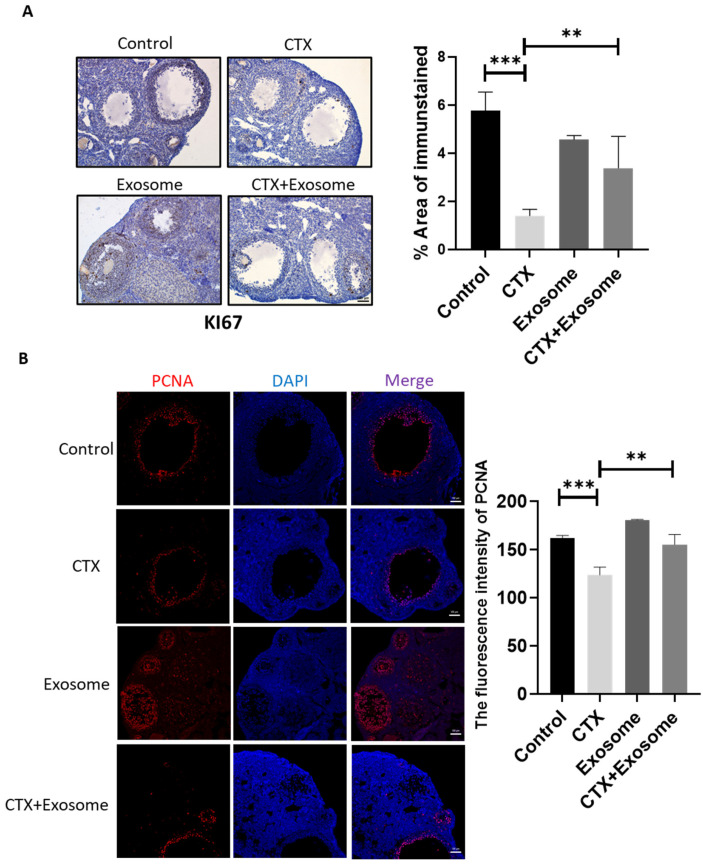
hiMSC exosomes promoted the proliferation of ovarian granulosa cells. (**A**) The expression level of KI67 was upregulated in the CTX + exosome group as compared to the CTX group by immunohistochemistry examination. (**B**) The expression level of PCNA upregulated in the CTX + exosome group as compared to the CTX group by immunofluorescence assay. Scale bars = 100 μm. ** *p* < 0.01, *** *p* < 0.001.

**Figure 7 molecules-28-02112-f007:**
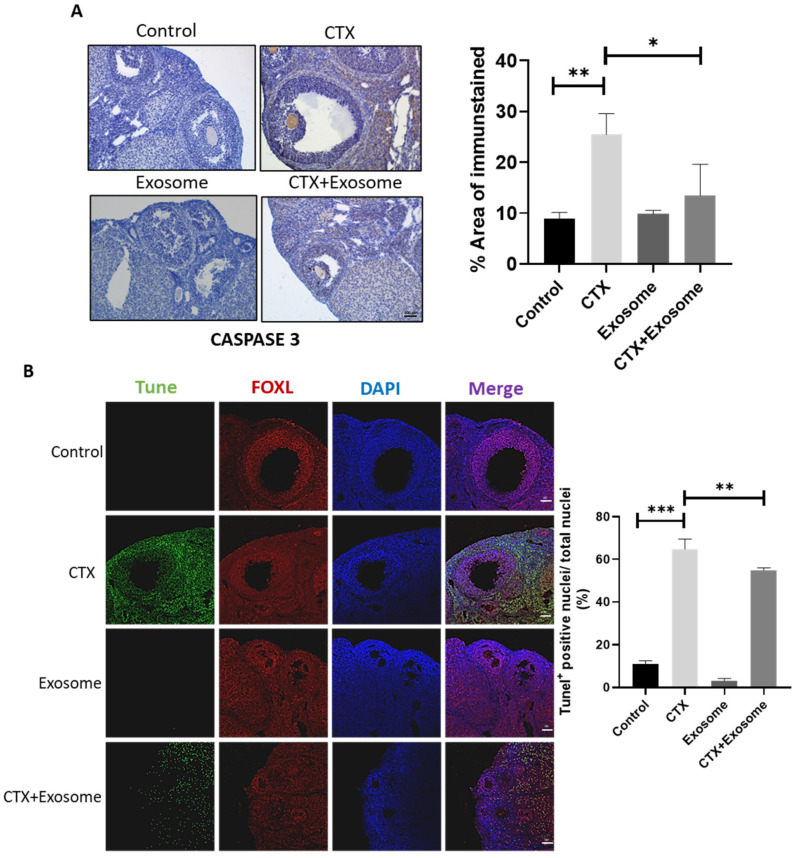
hiMSC exosomes inhibited ovarian granulosa cell apoptosis. (**A**) The level of CASPASE 3 was decreased in the CTX + exosome group when compared to the CTX group by immunohistochemistry examination. (**B**) Apoptosis was examined by TUNEL staining. Apoptotic cells were stained (in green), ovarian granulosa cells were stained with FOXL2 (in red) and cell nuclei were stained with DAPI (in blue). Scale bars = 100 μm. * *p* < 0.05, ** *p* < 0.01, *** *p* < 0.001.

**Figure 8 molecules-28-02112-f008:**
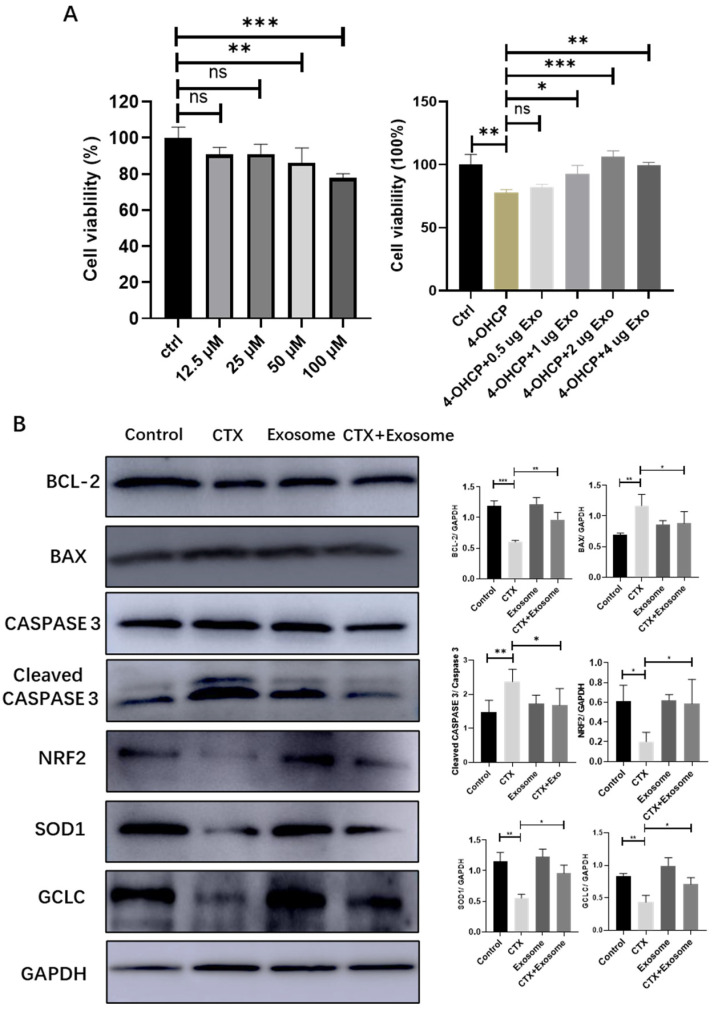
hiMSC exosomes reduced ovarian granulosa cellular oxidative stress. (**A**) MTT assay confirmed that hiMSC exosomes promoted the proliferation of mouse ovarian granulosa cells. (**B**) Western blotting results showed the expression of proteins related to apoptosis, and both NRF2 and proteins related to oxidative stress in granulosa cells are shown. * *p* < 0.05, ** *p* < 0.01, *** *p* < 0.001; ns, no significance.

## Data Availability

Not applicable.

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
