# Peer review of "Human Pluripotent Stem Cell–Mesenchymal Stem Cell-Derived Exosomes Promote Ovarian Granulosa Cell Proliferation and Attenuate Cell Apoptosis Induced by Cyclophosphamide in a POI-like Mouse Model"

_molecules, 2023, doi:10.3390/molecules28052112_

Round 1

Reviewer 1 Report

In this study, Zhang et al. investigated the roles of exosomes (Exos) in the growth and death of ovarian granulosa cells in a premature ovarian insufficiency (POI)-like mouse model treated by cyclophosphamide (CTX). This study was well-designed, and solid data supported the conclusions. I have the following suggestions for the authors to improve the manuscript.

[1] Since the addition of Exos promoted granulosa cell proliferation, increased ovary weight, and decreased apoptosis in the presence of CTX, the concept of Exos promoting cell growth (contributed by proliferation and cell survival) should be added to the manuscript title.

[2] Apoptosis can be measured by measuring the expression of cleaved caspase 3 but not caspase 3. Figure 7 shows the staining of caspase 3 in tissues; the related observations cannot be used to conclude apoptosis. Cleaved caspase 3 stainings should be used in this experiment. . Since caspase 3 (Figure 7) and cleaved caspase 3 (Figure 8) have been mentioned in this manuscript, was cleaved caspase 3 actually used in Figure 7?

[3] Quantitative results of Figures 6 and 7 should be provided to support the related conclusions.

[4] In the Results section, a rationale should be provided for each experiment. The authors can use 1-2 sentences to describe the reason(s) for each assay. Furthermore, it is better to describe the results using quantitative language, for example, "increased by xx%."

[5] The authors are encouraged to execute mechanistic studies to make this study more meaningful.

[6] The Conclusion section should discuss the limitations of this study, future studies, and the significance/implications/application of this study.

[7] This manuscript should be improved by applying the rules of scientific writing, for example, avoiding using uncommon abbreviations in the manuscript title and the list of keywords.

Reviewer 2 Report

The manuscript entitled " hiMSC-derived exosomes attenuate ovarian granulosa cell 2 apoptosis induced by cyclophosphamide in POI-like mouse 3 model " has been reviewed, and the following opinions were concluded:

The study has assessed hMSC-derived exosomes' effect on a POI-like mouse model. An adequate number of tests, such as hMSC quality assessment, exosome quality assessment, hormonal assays, staining experiments, viability-related assays, and animal fertility assays, have been performed to support the idea that MSC-derived exosomes are effective in ameliorating POI. The provided figures are of high quality, and the subjects are well ordered.

Despite all the efforts, the study can be improved further as well.

1. The text should be wholly revised in terms of punctuation, grammar, and fluency, preferably by a native English speaker.

2. The terms "at libitum" in lines 334 and 335 and "intralbursa" in line 325 should be addressed.

3. The term "hiMSC" seems to be misleading. It is suggested to use the full term in the title. Additionally, it is suggested to replace "human mesenchymal stem cells" with "hMSCs" throughout the whole text for better integrity and similarity to other available published texts.

4. The term "exosomes" is not a lengthy word and thus, does not need to be abbreviated. "hMSC-EXOs" can be utilized either.

5. The subtitle "2.2. Exos were collected from the cultured hiMSCs" could be stated as "Collection of exosomes from hMSCs."

6. "PBS group" should be replaced with "control group".

7. In Section 4.1, the provider of surface marker detection kits should be mentioned. It is better to mention the light microscope model and magnification power as well.

8. A separate abbreviation section should be considered as well.

9. It is proposed to add the limitations of the study to the discussion section.
